# Disability of Dialysis Patients and the Condition of Blood Vessels

**DOI:** 10.3390/jcm9061806

**Published:** 2020-06-10

**Authors:** Tomasz Gołębiowski, Mariusz Kusztal, Andrzej Konieczny, Krzysztof Letachowicz, Ada Gawryś, Beata Skolimowska, Bożena Ostrowska, Sławomir Zmonarski, Dariusz Janczak, Magdalena Krajewska

**Affiliations:** 1Department of Nephrology and Transplantation Medicine, Wroclaw Medical University, 50-556 Wroclaw, Poland; mariusz.kusztal@umed.wroc.pl (M.K.); andrzej.konieczny@umed.wroc.pl (A.K.); krzysztof.letachowicz@umed.wroc.pl (K.L.); adagawrys@gmail.com (A.G.); slawomir.zmonarski@umed.wroc.pl (S.Z.); magdalena.krajewska@umed.wroc.pl (M.K.); 2Department of Occupational Therapy, University School of Physical Education in Wroclaw, 51-612 Wroclaw, Poland; beata.skolimowska@awf.wroc.pl (B.S.); bozena.ostrowska@awf.wroc.pl (B.O.); 3Department of Vascular, General and Transplantation Surgery, Wroclaw Medical University, 50-556 Wroclaw, Poland; dariusz.janczak@umed.wroc.pl

**Keywords:** hemodialysis, disability, end-stage renal disease, arterial stiffness, pulse wave velocity

## Abstract

The number of elderly hemodialysis patients is constantly increasing worldwide. This population has a high burden of comorbid conditions, which impair daily activities. The aim of the study was to analyze problems of disability in hemodialysis patients in the context of cardiovascular (CV) complications and vascular condition. In this cross-sectional study, 129 patients (mean age 64.5) were enrolled. The degree of disability in Barthel index (Bi) and 10-year cardiovascular risk (QRISK^®^3) were assessed. A Mobil-O-Graph monitor was used for measuring hemodynamic parameters. Only 6.2% of patients were professionally active, 19% used a wheelchair for transport, and 16% used crutches. More than half (51%) were independent in everyday activities reaching 80–100 points on Bi. The rest, with Bi < 80, were considered as dependent. The most common causes of disability were CV complications. The independent group (80–100 points) was characterized by significantly lower pulse wave velocity (PWV) and lower QRISK^®^3 compared to dependent patients. The degree of disability negatively correlated with age, PWV, and QRISK^®^3. Multivariate logistic regression revealed that disability (Bi < 80) was independently associated with CV events in the past adjusted odds ratio (adj.OR) 4.83 (95% confidence interval (95% CI): 1.74–13.41) and higher PWV adj.OR 1.45 (95% CI: 1.15–1.82). Our results indicate that CV diseases are the most important cause of functional impairment.

## 1. Introduction

Patients aged 75 years or older represent the fastest growing segment of the subjects starting dialysis [1]. This population has a high burden of comorbid conditions, including cardiovascular (CV) diseases, which lead to disability and the need for continuous care resulting from problems in performing activities of daily living (ADLs). In this group, ADLs disability is strong and independent predictor of mortality [2]. The etiology of disability is multifactorial and may be divided into two groups, namely those associated with atherosclerosis and non-atherosclerotic diseases. In hemodialysis (HD) patients, atherosclerosis progresses much faster than in age-matched patients, with a normal glomerular filtration rate, and it is an important risk factor of cardiovascular (CV) events [3]. Heart failure, hemiplegia after stroke, or lower leg amputation are only examples of many arteriosclerotic consequences, which may limit the ability to perform daily self-care tasks. However, some data indicate that the musculoskeletal system involvement may significantly decrease the physical function of patients with end-stage renal disease [4]. In many cases, both CV diseases and disturbances of the musculoskeletal system may coexist, leading to extend functional disability (FD). In such circumstances, it is difficult to judge which illness has a more important impact on limiting daily activities. Additionally, disturbances and symptoms related to dialysis like sarcopenia, depression, cognitive impairment, visual impairment, and degenerative joints disease may aggravate or even be a leading reason for FD [5]. Pulse wave velocity (PWV) is generally regarded as surrogate to large/medium arterial stiffness and it is a common predictor of CV events in elderly on-dialysis patients. The association between disability due to CV diseases and PWV is not well studied. We hypothesized that the leading cause of functional impairment are cardiovascular complications, as a consequence of changes in vascular system.

The aim of the study was to analyze the problems of FD in HD patients in the context of cardiovascular complications and vascular condition.

## 2. Material and Methods

The potential participants were identified through a review of all patients undergoing hemodialysis who were recruited from two dialysis stations.

The inclusion criteria for hemodialysis patients were as follows: (1) age 18 years and over, (2) received regular dialysis for at least 6 months, and (3) were able to provide informed consent. The exclusion criteria included the following: (1) currently enrolled in another study or (2) receiving emergency in-patient care within four weeks. Written informed consent was received from each patient entering the study. Of the 139 eligible participants, 10 did not agree to take part in the study. Patients who were not recruited were similar in age and gender to those participating in the study. 

One hundred twenty-nine patients (mean age 64.5, 54 females, with mean dialysis vintage 61 months) were enrolled in this cross-sectional study. Demographic and comorbidity data were collected from medical records and from a direct interview. This information allowed us to assess their degree of disability (Barthel index) and a 10-year cardiovascular risk (QRISK^®^3 online calculator).

Cardiovascular (CV) diseases included diseases of the heart, vascular diseases of the brain, and diseases of blood vessels [6] particularly: myocardial infarction; heart failure New York Heart Association (NYHA) class > II, atrial fibrillation; implantable medical devices (IMD), i.e., cardio-stimulator or cardioverter-defibrillator; ischemic or hemorrhagic stroke; amputation due to extremity ischemia or diabetic foot; and surgery to repair an aortic aneurysm. 

Coronary heart disease was defined as self-reported or on the basis of a history of myocardial infarction, coronary angioplasty, or bay-pass grafting. Stroke included a history of transient ischemic attack in the past or an ischemic/hemorrhagic event with neurological consequences. Peripheral arterial disease was defined as self-reported or medical report history of lower extremity angioplasty or significant vessel stenoses in imagine tests, amputation due to ischemia. 

In most of patients (82 out of 129, 63%), two or more disorders were found, which may have a negative impact on performing ADLs. In such cases, an algorithm of choosing the leading disorder, responsible for disability, relies on determining which disorder was the most important in ADLs disability. Weakness associated with dialysis, as a leading disorder of disability, was found as self-reported when severely limiting the mobility and after excluding other potential factors of low muscle strength. Joint and skeletal system manifestation encompassed disorders typical for aging, i.e., degenerative joint diseases, but also disorders related to dialysis, such as following: Chronic Kidney Disease–Mineral and Bone Disorder (CKD-MBD), renal osteodystrophy due to secondary hyperparathyroidism, and dialysis-related amyloidosis.

The Barthel index is a 10-item instrument, measuring functional independence, in personal activities of daily living (ADLs) and the need for supervision or support. It is composed of 10 categories and includes the following items: feeding, bathing, grooming, dressing, bowel control, bladder control, toilet use, and transfers (bed to chair and back) [7,8]. The scale is a 100-point scale and is measured in increments of 5 points. The total score was 100 points. Subjects were categorized according to the Barthel index as follows: 80–100 points, independent including patients fully independent with 100 points; 60–79 points, minimally dependent; 40–59 points, partially dependent; 20–39 points, highly dependent; and <20 points, fully dependent.

QRISK^®^3 score is an online tool evaluating a 10-year risk of cardiovascular accident and it was validated for different ethnicities and CKD patients [9]. It takes into account the following factors: age, ethnicity, diabetes, hypertension, smoking, angina or heart attack in a 1st degree relative <60 y, atrial fibrillation, blood pressure treatment, blood pressure and chronic kidney disease, migraine, corticosteroids, systemic lupus erythematosus, atypical antipsychotics medication, severe mental illness, and erectile dysfunction. The results are presented as a risk of a heart attack or stroke within the next 10 years.

In addition, ambulatory measurements of hemodynamic parameters with Mobil-O-Graph monitor (Industrielle Entwicklung Medizintechnik und Vertriebsgesellschaft GmbH (IEM), Stolberg, Germany), which records oscillometric arm blood pressure: systolic and diastolic blood pressure (SBP and DBP), central systolic and diastolic blood pressure (cSBP and cDBP), pulse pressure (pPP), ejection fraction (EF), cardiac output (CO), and pulse waves. It calculates the augmentation index normalized to the heart rate (Aix@75) as a measure of wave reflections, and pulse wave velocity (PWV) as a measure of arterial stiffness. All tests were performed before the start of the one-day hemodialysis sessions.

Statistical analysis was performed using standard software (Statistica Version 13.3, (StatSoft, Tulsa, OK, USA)). Data for continuous variables were expressed as means and standard deviations. Paired independent sample *t*-test was used to compare the means of two continuous variables and the chi-square test for categorical variables. The relationship between two continuous variables was evaluated using the Pearson correlation. The unadjusted and adjusted multivariate Cox regression analysis was presented as an odds ratio (OR; 95% confidence intervals (CI)). A *p*-value < 0.05 was considered significant.

Ethics approval was granted by the Ethics Board of Wroclaw Medical University No KB 587/2018.

## 3. Results

The baseline demographic and blood chemistry characteristic are displayed in Table 1. Diabetes and renovascular disease were the most common causes of kidney failure, followed by others. Ischemic nephropathy was a more frequent cause of end-stage renal disease in dependent and glomerulonephritis dominate in group of independent patients (Table 2). In the study group of 129 patients (pts.), only 8 (6.2%) were professionally active, 40 (31%) received disability pension, and 81 (62.8%) were retired. Twenty-five patients (19%) used a wheelchair for transport and 21 (16%) used crutches. More than half (51%) were independent in everyday activities, reaching 80–100 points on the Barthel index; however, only 14% were fully independent (100 points). Minimally dependent (60–79 points) constituted 27% (35 pts) of all patients, partially dependent (40–59 points) constituted 10% (13 pts), highly dependent (20–39 points) constituted 7% (9 pts), and fully dependent (<20 points) constituted 5% (6 pts). The most common causes of disability were heart problems, including heart failure in 20.9% of the patients, lower limb ischemia, diabetic foot, and amputation in 13.2%, and cerebrovascular event in 10.9%, followed by weakness associated with dialysis and degenerative joints disease. In the dependent group, cardiovascular diseases of heart, brain, and blood vessels as a leading cause of disability were more frequently observed in comparison to the independent group of patients. In contrary symptoms related to uremia (i.e., sarcopenia), weakness associated with dialysis was dominant in the independent subjects (Table 3).

The group of independent individuals (80–100 points on the Barthel index) was characterized by a significantly lower PWV, QRISK^®^3, and fewer CV events in the past compared to the dependent group (Table 1). In the group of independent patients, a few more patients received antihypertensive treatment (ꞵ-blockers, α-blockers, calcium channel blockers, angiotensin converting enzyme inhibitors, angiotensin receptor blockers were recorded) but the chi-square test did not show significant difference compared to the group of dependent patients (*p* > 0.05). The degree of disability according to Barthel index was negatively correlated with age, PWV, and QRISK^®^3 (Figure 1).

Development and results of logistic regression model.

Crude odd ratios (ORs) for all predictors are shown in Table 4. The factors associated with having ADL disability (Barthel index < 80 points) included cardiovascular events in the past, higher QRISK^®^3, age, higher PWW, lower diastolic pressure (DBP and cDBP), higher central pulse pressure (cpPP). No significant relationship with disability was observed for Aix@75, BMI, dialysis vintage SBP and MBP, and parameters of heart function (EF, CO).

Based on a correlation analysis between each predictor, the groups of strongly associated predictors were extracted and finally one representative was chosen to avoid collinearity. All the remaining predictors were entered into a multivariable logistic regression model to obtain adjusted estimates. CV events in the past and higher PWV were the strongest, independent predictors for ADL disability with adjusted ORs 4.83 (95% CI: 1.74–13.41) and OR 1.45 (95% CI: 1.15–1.82), respectively (Table 5).

## 4. Discussion

Disability is defined as difficulty or dependency in carrying out activities that are essential to independent living. The Barthel index encompasses 10 activities of daily living (ADLs): feeding, bathing, grooming, dressing, using a toilet, bowels and bladder incontinence, transfers, mobility, and use of stairs. The inability to perform such activities engages family members or other informal caregivers. In addition, ADLs disability may strongly predict morbidity, mortality, and hospitalization in dialysis adult [10].

The results of our recent study reflect three important findings regarding ADLs disability. The first was that disability is common in the end-stage renal disease (ESRD) population. Half of the studied population (48% of patients) was functionally dependent and their Barthel index was lower than <80 points, which means that they need support in at least one ADL. Previous studies reported the approximate rates of functional impairment in ADLs (29–52%) [11,12]. The prevalence of disability among patients with chronic kidney disease (CKD), not receiving dialysis, accounts for 18–24% of elderly patients over 65 years. In Kavanaghetet al.’s study, even 58.8% of the participants demonstrated dependence in at least one ADL or instrumental activities of daily livings (IADLs) [13]. Consistent results were reported by Cook et al., who recruited 162 patients aged 65 and older (mean age was 75 years). In this cohort, 52% of subjects were dependent both in ADLs and IADLs [14]. These differences may be explained by the different age of the study population. Only 14% of the studied dialysis population had no functional impairment of any type. The approximately similar percentage (5%) was observed in the aforementioned study conducted by Cook et al. [14]. Characteristics of disabled patients overlap with the frailty status, which is defined as the presence of three or more, out of five, of following criteria: unintentional weight loss, weakness, self-reported fatigue, slow walking speed, and low physical activity level. Frailty status is a well-known CV risk factor in ESRD and in the non-renal population [15,16]. 

The second discovery was that cardiovascular (CV) complications are the leading cause of disability. The history of CV event was observed in 66% of study subjects and more frequently in the group of dependent patients (89% vs. 45%; Table 1). CV events in the past were also the strongest predictor of functional disability (FD), increasing the risk of disability almost five times (Table 5). Heart failure as a complication of coronary heart disease was the most frequent cause of FD (in 20.9% of pts.), followed by other complications associated with arteriosclerosis, i.e., peripheral arteries in the lower extremity (including amputation) and stroke (Table 3). Data from other authors confirm that CV diseases are the main cause of death both in community-dwelling elderly and patients with reduced kidney function, but coronary artery disease and stroke are the two most important causes of death [15,17]. 

The third important finding, derived from our study, was that there is a significant association between vascular changes, expressed by arterial stiffness, and the degree of ADL disability in the Barthel index (Bi). The group of dependent patients is characterized by significantly increased arterial stiffness expressed by pulse wave velocity (PWV; Table 1). Lower disability score (Bi) was associated with higher PWV, which was an independent predictor for dependency status, with an increase in adjusted odds ratio (adj.OR) 1.4 pro 1 m/s of PWV. In addition, we found lower diastolic blood pressure and higher central pulse pressure in the group of dependent patients (Table 1). This is in line with higher PWV score, a well-known surrogate of large arterial stiffness, and strongly suggests higher arterial stiffness. To our knowledge, it is the first study evaluating the relationship between PWV and disability score. The mechanism underlining arterial stiffness in the CKD population is complex and can be attributed to the progression of vascular calcification, owing to Chronic Kidney Disease–Mineral and Bone Disorder (CKD-MBD) and an imbalance between factors promoting calcification i.e., phosphate, parathormone, fibroblast growth factor-23 (FGF−23), and calcification inhibitors (e.g., fetuin-A) [18]. Faster progression of increased aortic PWV can be found more often in the CKD population than among individuals with normal kidney function [19]. A meta-analysis of prospective observational data confirms that PWV was a significant predictor of coronary heart disease, stroke, and cardiovascular events, with hazard ratios of 1.23, 1.28, and 1.30, respectively [20]. Regardless, PWV is a useful, non-invasive, diagnostic and prognostic tool for the prediction of cardiovascular disease in the CKD population; however as a mortality predictor, it may not necessary show superior usefulness comparing to other validated clinical risk scores. Recent comparison of PWV vs. annualized rate of occurrence (ARO) risk score in patients with ESRD indicate that PWV has inferior prognostic power for all-cause and cardiovascular mortality compared to ARO risk score [21,22]. Additional data point that 24-h PWV may have a higher prediction value than PWV in the office setting for all-cause-mortality in hemodialysis patients [23]. The 2018 European Society of Cardiology Guidelines determine the PWV threshold of 10 m/s as suggestive of increased cardiovascular risk and is appropriate to be utilized to stratify intermediate risk patients (Grade 2b recommendation) [24]. Aortic PWV was also broadly used in interventional studies [25,26].

Data from the Dialysis Outcomes and Practice Patterns Study (DOPPS study) confirm that there is an independent relationship between functional impairment and adverse health outcomes [27]. The accumulation of conventional and unconventional risk factors, typical for CKD patients, initiate the pathological chain process, which includes the following elements: increased arterial stiffness, hypertension, left ventricular heart hypertrophy, arteriosclerosis with cardiovascular complications, including vascular bed in the brain, heart and peripheral arteries. The final element is disability or death. Owing to non-traditional risk factors (including biochemical abnormalities of hyperphosphatemia and hyperparathyroidism, renal osteodystrophy, and vascular calcification), the risk of in-hospital FD progression after CV event is higher in dialysis than in non-dialysis patients [28]. This pathway links functional dependence with changes in the vascular system and seems to be crucial in the dialysis population.

Naturally, cardiovascular diseases are not a separate culprit of impaired functioning. In 22% and 14% of participants, fatigue/weakness associated with dialysis and skeletal system manifestations were the leading cause of FD, respectively. This issue should be discussed in connection, as both include non-atherosclerosis disorders frequently limiting functional independence. Sarcopenia is a common disturbance in the dialysis population and is associated with physical disability and mortality [29,30]. The etiology is multifactorial, although a sedentary lifestyle is regarded as the most important factor of low physical fitness [31]. In addition, many other factors may contribute to this disturbance: anemia, the uremic milieu itself, inflammatory state, and malnutrition [32,33]. General weakness was noted in 60% of patients with secondary hyperparathyroidism prepared to undergo a parathyroidectomy [34]. In the Dialysis Morbidity and Mortality Study (DMMS), out of 2275 dialysis patients, 66% were described as weak and fragile [35] in comparison the group of people over 65 years of age, without kidney disease, this trait was found in 6.9% [36] to 16.3% [37]. Renal osteodystrophy (ROD) is common in CKD and two types are encountered: high turnover and low turnover disease. The prevalence of high turnover bone disease (osteitis fibrosa cystica) among dialysis patients has markedly decreased [38,39]. The symptomatology of these two different disorders includes changes in bones, joints, and the vessels. In a study by Afifi et al., musculoskeletal manifestations of ROD were frequently observed in the dialysis population. Arthralgia was present in 83% of the subjects, subperiosteal resorption of the terminal phalanges and achilles tendinopathy in 67.9%. These abnormalities corresponded with a low self-report functional disability index, as shown in the health assessment questionnaire (HAQ-DI) [4].

In our analysis, the duration of renal replacement therapy has no impact on functional impairment. Due to the cross-sectional character of the study, this finding should be regarded with caution, as from clinical practice functional decline is usually observed in most patients after years of dialysis. In a large study from Taiwan, using the Barthel index for a longitudinal evaluation of functional dependence, in follow-up time of 13 years, the authors found that patients who started dialysis at an age above 35 years experienced approximately 3 years of disability; however, with age increases the proportion of functional disabilities and care needs [40]. It should be also noted that FD in elderly patients is exacerbated and frequently observed after the initiation of hemodialysis treatment [41].

Identification of a functional decline in the HD population may be clinically important, because early intervention may diminish the degree of disability. Some evidence suggests that multidisciplinary rehabilitation programs may improve the ability to perform different tasks of daily living. These programs should encompass counselling with a specialized team, including a physiotherapist, psychiatrist, behavioral specialist, geriatrist, or others when appropriate. Individualized, supervised, or grouped exercise training may significantly improve muscle strength and performing ADLs even in low functioning patients [42,43]. Some limited evidence studies suggest that exercise is a potential strategy to improve the vascular condition and lower blood pressure in CKD [44]. Increasing physical activity levels should be a major goal at all levels of health care [45]. Exercise is hypothesized to minimize overall cardiovascular risk and potentially mitigates the impact of hypertension on arterial stiffness. One cross-over study of 19 patients showed a trend towards improvement in PWV in patients who undertook interdialytic exercise, with subsequent deterioration in PWV after cessation of exercise [46], but another study has failed to demonstrate that exercise has any significant impact on PWV [47]. 

A few limitations should be underlined when assessing our findings. Precise data indicate that except stiffness of large and small arteries, many factors may influence hemodynamic parameters of pulse wave analysis including cardiac output, existence of arteriovenous fistula, and hydration status of the individual patients [48]. Owing to the cross-sectional study design, we were unable to comment on the dynamic nature of disability over time. There might be a subjective element in the assessment of the main cause of disability, because in a significant number of patients two or more disturbances, diseases, or symptoms coexisted and only most the important one was chosen. Sarcopenia or weakness associated with dialysis may be confused with depression or sleep disturbances, as they all appear as fatigue. Our data are also limited by the fact that we recruited dialysis patients from only two in-center HD units.

## 5. Conclusions

To our knowledge, this is first study evaluating functional disability (FD) in hemodialysis (HD) patients in the context of vascular conditions. Our results indicate that half of HD patients show some degree of FD, with considerable consequences on their autonomy. Functional independence in the HD population appears to be the exception rather than the rule. Cardiovascular (CV) events are the most important cause of functional decline. The group of patients with a higher degree of disability is usually older and characterized by a high burden of CV complications. Additionally, more advanced changes in the arteries might be the reason for the higher risk of CV incidents in the future.

## Figures and Tables

**Figure 1 jcm-09-01806-f001:**
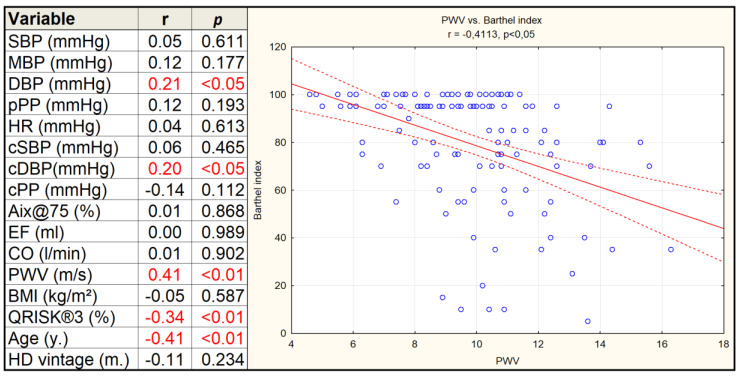
Correlations between Barthel index and other variables. r-index displayed in red for statistically significant correlations *p* < 0.05. Abbreviations: SBP, systolic blood pressure; MBP, mean blood pressure; DBP, diastolic blood pressure; pPP, pulse pressure; cSBP, central systolic blood pressure; cDBP, central diastolic blood pressure; cpPP, central pulse pressure; Aix@75, augmentation index normalized with 75/minute heart rate; EF, ejection fraction; CO, cardiac output; PWV, pulse wave velocity; BMI, body mass index; QRISK^®^3, 10 year risk of cardiovascular accident; HD, hemodialysis.

**Table 1 jcm-09-01806-t001:** Baseline characteristics of the groups: independent (80–100 points on the Barthel index) and dependent patients (<80 points on the Barthel index).

	Independent (*n* = 67)		Dependent (*n* = 62)		
Blood pressure and HR	Mean	±	Mean	±	*p*-value
SBP (mmHg)	142.8	26.3	139.7	29.9	0.530
MBP (mmHg)	112.5	20.0	106.2	20.8	0.083
DBP (mmHg)	86.9	17.0	77.8	15.7	<0.05
pPP (mmHg)	55.9	17.5	61.9	20.9	0.078
cSBP (mmHg)	127.8	22.6	123.8	24.8	0.341
cDBP (mmHg)	89.0	17.5	79.6	16.0	<0.05
cPP (mmHg)	38.7	12.7	44.2	15.2	<0.05
HR (1/min.)	73.5	14.3	72.1	11.6	0.541
Hemodynamic					
EF (mL)	72.9	14.9	73.3	14.4	0.898
CO (L/min.)	5.3	1.1	5.2	1.1	0.869
TVR (s × mmHg/mL)	1.3	0.3	1.3	0.4	0.251
Cardiac index (L/min. × 1/m^2^)	2.9	0.7	2.9	0.8	0.677
Vascular stiffness					
PWV (m/s)	8.8	2.0	10.9	2.2	<0.05
Aix@75 (%)	21.6	14.2	22.9	12.9	0.602
Augmentation pressure (mmHg)	10.0	8.4	12.6	9.6	0.102
Reflex (%)	60.6	10.8	63.0	9.9	0.197
Sex, Age, and body size					
Female. No. (%) *	28 (41.8)		26 (41.9)		0.992
Age (y.)	57.9	14.7	71.6	12.5	<0.05
BMI (m^2^/kg^2^)	26.4	5.3	26.5	5.2	0.862
Weight (kg)	75.4	19.2	73.1	16.0	0.458
Cardiovascular					
QRISK^®^3 (%)	22.6	13.0	32.9	11.8	<0.05
History of CV events: No (%) *	30 (45)		55 (89)		<0.05
Antihypertensive agents					
ꞵ-blockers No (%) *	50 (74.6)		42 (67.7)		0.723
CCB No (%) *	46 (68.7)		26 (41.9)		0.102
ACEI/ABR No (%) *	7 (10.4)		5 (8.0)		0.671
Diuretics No (%) *	14 (20.9)		15 (24.2)		0.722
α-blockers No (%) *	20 (29.9)		20 (32.3)		0.830

Abbreviations: SBP, systolic blood pressure; MBP, mean blood pressure; DBP, diastolic blood pressure; pPP, pulse pressure; cSBP, central systolic blood pressure; cDBP, central diastolic blood pressure; cpPP, central pulse pressure; EF, ejection fraction; CO, cardiac output; TVR, total vascular resistance; PWV, pulse wave velocity; Aix@75, augmentation index normalized with 75/minute heart rate; BMI, body mass index; QRISK^®^3 score, 10 year risk of cardiovascular accident; CCB, calcium channel blocker; ACEI, angiotensin converting enzyme inhibitor; ARB, angiotensin receptor blocker; HR, heart rate; CV, cardiovascular; * chi-square test.

**Table 2 jcm-09-01806-t002:** Causes of renal failure.

Cause of Renal Failure	All Patients (*n* = 129)	Independent (*n* = 67)	Dependent (*n* = 62)	*p*-Value *
Ischemic nephropathy/hypertension No (%)	45 (34.9)	13 (19.4)	32 (51.6)	<0.01
Diabetic nephropathy No (%)	34 (26.4)	16 (23.9)	18 (29.0)	0.613
Glomerulonephritis No (%)	24 (18.6)	19 (28.4)	5 (8.1)	<0.05
ADPKD No (%)	8 (6.2)	6 (9.0)	2 (3.2)	0.205
Pyelonephritis (reflux, stones) No (%)	9 (6.98)	7 (10.4)	2 (3.2)	0.133
Urologic cancer (kidney, prostate) No (%)	6 (4.65)	4 (6.0)	2 (3.2)	0.480
Unknown No (%)	2 (1.55)	1 (1.5)	1 (1.6)	0.956
Myeloma No (%)	1 (0.78)	1 (1.5)	0 (0)	0.337

Abbreviations: ADPKD, Autosomal dominant polycystic kidney disease. * chi-square test between dependent and independent group.

**Table 3 jcm-09-01806-t003:** Disorders leading to disability.

Disorders Leading to Disability	All Patients (*n* = 129)	Independent (*n* = 67)	Dependent (*n* = 62)	*p*-Value *
Heart failure/coronary heart disease/acquired valvular heart disease	27 (20.9)	3 (4.5)	24 (38.7)	<0.01
Lower limb ischemia, diabetic foot and amputation	17 (13.2)	4 (6.0)	13 (21.0)	<0.05
Central nervous system disorder	14 (10.9)	3 (4.5)	11 (17.7)	<0.05
Sarcopenia/weakness associated with dialysis	28 (21.7)	26 (38.8)	2 (3.2)	<0.01
Join and skeletal system manifestations	18 (14.0)	9 (13.4)	9 (14.5)	0.878
Vision and hearing loss	4 (3.1)	2 (3.0)	2 (3.2)	0.939
Respiratory tract disorders	3 (2.3)	2 (3.0)	1 (1.6)	0.614
Fully independent	18 (14.0)	18 (26.9)	0 (0)	<0.01

* chi-square test between dependent and independent group.

**Table 4 jcm-09-01806-t004:** Univariate logistic regression. Variables correlated to dependent status (Barthel index < 80 points).

Variable	Estimate	OR	95% CI		*p*-Value
Age (year.)	0.075	1.08	1.04	1.11	<0.05
PWV (m/s)	0.499	1.65	1.33	2.04	<0.05
Aix@75 (%)	0.007	1.01	0.98	1.03	0.60
BMI (kg/m^2^)	0.006	1.01	0.94	1.08	0.86
QRISK^®^3 (%)	0.066	1.07	1.04	1.10	<0.05
HD (m)	0.004	1.00	1.00	1.01	0.15
SBP (mmHg)	−0.004	1.00	0.98	1.01	0.53
MBP (mmHg)	−0.015	0.98	0.97	1.00	0.09
DBP (mmHg)	−0.034	0.97	0.94	0.99	<0.05
pPP (mmHg)	0.017	1.02	1.00	1.04	0.08
HR (1/min)	−0.008	0.99	0.97	1.02	0.54
cSBP (mmHg)	−0.007	0.99	0.98	1.01	0.34
cDBP (mmHg)	−0.034	0.97	0.95	0.99	<0.05
cpPP (mmHg)	0.029	1.03	1.00	1.06	<0.05
EF (mL)	0.002	1.00	0.98	1.03	0.90
CO (mL/min.)	−0.003	1.00	0.97	1.03	0.87
History of CV (1)	2.271	9.69	3.85	24.37	<0.05

Abbreviations: OR, odds ratio; CI, confidence interval.

**Table 5 jcm-09-01806-t005:** Multivariate logistic regression.

Variable	Estimate	adj.OR *	95% CI		*p*-Value
History of CV (1)	1.576	4.83	1.74	13.41	<0.05
PWV (m/s)	0.370	1.45	1.15	1.82	<0.05
cDBP (mmHg)	−0.031	0.97	0.94	0.99	<0.05

Abbreviations: adj.OR, adjusted odds ratio. * correlated to Age (year), Aix@75, QRISK^®^3, MBP, cpPP.

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
