# Peer review of "Disability of Dialysis Patients and the Condition of Blood Vessels"

_jcm, 2020, doi:10.3390/jcm9061806_

Round 1
Reviewer 1 Report
Authors tried to demonstrate relationship between disability of dialysis patients and condition of blood vessels. There are several issues to be clarified and/or resolved as below,
1. Regarding the definition of “cardiovascular (CV) events” in this manuscript, I don’t agree the CV contains ischemic/hemorrhagic stroke. In my understanding, the stroke is cerebrovascular disease, except for cardiogenic stroke.
2. What the authors says in this paper seems to be no different from “PWV was significant predictor of coronary heart disease, stroke and cardiovascular events, with hazard ratios 1.23, 1.28, 1.30, respectively. PWV is regarded as useful, non-invasive, diagnostic and prognostic tool for assessing cardiovascular disease, in the CKD population.”, which is cited in this article. In short, the CV events, that is predicted by PWV, causes the decline of ADLs, or the decrease of Barthel index. In my thought, it is an obvious thing, so the conclusions of this paper are nothing new.
Author Response
Authors tried to demonstrate relationship between disability of dialysis patients and condition of blood vessels. There are several issues to be clarified and/or resolved as below,
- Regarding the definition of “cardiovascular (CV) events” in this manuscript, I don’t agree the CV contains ischemic/hemorrhagic stroke. In my understanding, the stroke is cerebrovascular disease, except for cardiogenic stroke.
Response: We understand this remark, however we used broad definition of cardiovascular diseases (CVDs) of Thomas, H.; Diamond, J.; Vieco, A.; Chaudhuri, S.; Shinnar, E.; Cromer, S.; Perel, P.; Mensah, G.A.; Narula, J.; Johnson, C.O., et al. Global Atlas of Cardiovascular Disease 2000-2016: The Path to Prevention and Control. Glob Heart 2018, 13, 143-163, doi:10.1016/j.gheart.2018.09.511
According to this guidelines CVDs include diseases of the heart, vascular diseases of the brain and diseases of blood vessels.
We added a reference and following sentence in Material and Methods section
“Cardiovascular (CV) diseases include diseases of the heart, vascular diseases of the brain and diseases of blood vessels [6] particularly:”
- What the authors says in this paper seems to be no different from “PWV was significant predictor of coronary heart disease, stroke and cardiovascular events, with hazard ratios 1.23, 1.28, 1.30, respectively. PWV is regarded as useful, non-invasive, diagnostic and prognostic tool for assessing cardiovascular disease, in the CKD population.”, which is cited in this article. In short, the CV events, that is predicted by PWV, causes the decline of ADLs, or the decrease of Barthel index. In my thought, it is an obvious thing, so the conclusions of this paper are nothing new.
Response: We thank you for this important comment. From our point of view CVDs end points (coronary heart disease, stroke etc.) are not similar with functional decline (FD). As we shown FD in CKD patients may be associated with many factors i.e. CV events, sarcopenia, depression, degenerative joint disease etc. Our study for the first time shown significant relationship between arterial stiffness surrogate, namely PWV and ADLs decline assessed by widely used in Poland Barthel Index. We could not find similar papers.
We also found that CV events in the past were more important culprit of FD than others factors related to uremia (post dialysis fatigue, sarcopenia, depression).
Reviewer 2 Report
The authors described the disability of dialysis patients and the condition of blood vessels in this manuscript. This manuscript is interesting, because the
The authors described the disability of dialysis patients and the condition of blood vessels in this manuscript. This manuscript is interesting, because the authors showed three important findings. Namely, the first is that disability is common in end-stage renal disease population, the second is that a leading cause of disability is cardiovascular complications, and the third is the significant association between vascular changes expressed by arterial stiffness and degree of activity of daily living. However, some concerns have been raised.
(1) Measurements of hemodynamic parameters change according to body fluid volume. The authors indicated that all tests were performed before scheduled dialysis sessions. However, it is not clear whether the tests were performed before the start of the one-day or two-day hemodialysis session. The authors should indicate the schedule clearly.
(2) It is important whether the patients took antihypertensive drugs to analyze and interpret this study. Therefore, the authors should indicate the medication status.
(3) Sex differences are not shown between independent and dependent patients. The authors should indicate them in Table 1.
(4) Differences of causes of renal failure are not shown between independent and dependent patients. The authors should indicate them in Table 2.
(5) Differences of disorders leading to disability are not shown between independent and dependent patients. The authors should indicate them in Table 3.
(6) P values are not indicated in Figure 1. The authors should add the data in Figure 1
(7) I think that the incident rates of cardiovascular event were 44 % (30/67) in independent patients and 89 % (55/62) in dependent patients. If so, the authors should revise the data in Table 1 and page 9, 3rd paragraph.
(8) The frequency of heart failure is 20.93% according to Table 3. However, the authors indicated the frequency as 27% in page 9, 3rd paragraph. The authors should revise the mistake.
(9) There are some typographical errors in the manuscript. For example, join is not correct and joint is correct. FD is an abbreviation for functional disability but not functional decline (page 9, 3rd paragraph). The authors should correct the errors adequately.
Author Response
The authors described the disability of dialysis patients and the condition of blood vessels in this manuscript. This manuscript is interesting, because the authors showed three important findings. Namely, the first is that disability is common in end-stage renal disease population, the second is that a leading cause of disability is cardiovascular complications, and the third is the significant association between vascular changes expressed by arterial stiffness and degree of activity of daily living. However, some concerns have been raised.
Response: We thank the reviewer for the kind words. We have revised the manuscript accordingly, and hope that additional informations supplied to the text make it more valuable .
(1) Measurements of hemodynamic parameters change according to body fluid volume. The authors indicated that all tests were performed before scheduled dialysis sessions. However, it is not clear whether the tests were performed before the start of the one-day or two-day hemodialysis session. The authors should indicate the schedule clearly.
Response: We thank the reviewer for this remark. We agree that hemodynamic parameters may change depending hydration status of the individual patients that is why all measurements were performed before the start of the one-day hemodialysis session. This information was added to the revised manuscript in Material and Methods section. The sentence
“All tests were performed before scheduled dialysis sessions.”
was replaced with
“All tests were performed before the start of the one-day hemodialysis session.”
(2) It is important whether the patients took antihypertensive drugs to analyze and interpret this study. Therefore, the authors should indicate the medication status.
Response: We thank the reviewer for this advice and we agree that medication status should be replenished as important part patient’s hemodynamic status, because differences related to demand for antihypertensive agents (AHA) in dependent and independent group may add new interesting information. We looked again for any medication used by patients in both group as was suggested. Only few more patients in “independence” group received antihypertensive drugs (êžµ-blockers, α-blockers, calcium channel blocker, ACEI/ARB were counted) however it was not statistically significant. This results however shows a trend for higher AHA demand in more heathy independent patients. The Table 1 was accordingly changed and the following information were included in Results section.
“In the group of independent patients few more patients received antihypertensive treatment (êžµ-blockers, α -blockers, calcium channel blockers, angiotensin converting enzyme inhibitors, angiotensin receptor blockers were recorded)but the Chi-square test did not show significant difference to group of dependent patients (p>0.05).”
(3) Sex differences are not shown between independent and dependent patients. The authors should indicate them in Table 1.
Response: We thank the reviewer for this remark. The Table 1 was naturally changed as above.There were no differences in regard of sex in both groups.
(4) Differences of causes of renal failure are not shown between independent and dependent patients. The authors should indicate them in Table 2.
(5) Differences of disorders leading to disability are not shown between independent and dependent patients. The authors should indicate them in Table 3.
Response referred to point (4) and (5): We thank the reviewer for those two advices. It is naturally important issue to study differences in both groups. We have replenished the Table 2 and 3. The results indicate that ischemic nephropathy/hypertension is more frequent cause of renal failure in dependent group and glomerulonephritisdominate in group of independent patients. Such additional statement was added to revised manuscript – Results section.
Ischemic nephropathy was more frequent cause of end-stage renal disease in dependent and glomerulonephritis dominate in group of independent patients.
The results in Table 3 indicate that CV diseases (Heart failure/coronary heart disease/acquired valvular heart disease and lower limb ischemia/diabetic foot/amputation and central nervous system disorder) as a leading cause of disability were more frequently observed in dependent group. In contrary sarcopenia/weakness associated with dialysis dominate in independent group of patients.This differences were statistically significantly.
Additional statement was added to revised manuscript – Results section.
In dependent group cardiovascular diseases of heart, brain and blood vessels as a leading cause of disability were more frequently observed in comparison to independent group of patients. In contrary symptoms related to uremia i.e. sarcopenia, weakness associated with dialysis dominated in independent subjects.
(6) P values are not indicated in Figure 1. The authors should add the data in Figure 1
Response: We thank the reviewer for this suggestion. To accommodate the reviewer, we have added p values in Figure 1.
(7) I think that the incident rates of cardiovascular event were 44 % (30/67) in independent patients and 89 % (55/62) in dependent patients. If so, the authors should revise the data in Table 1 and page 9, 3rd paragraph.
Response: We thank the reviewer for founding of this mathematical error. We are sorry and naturally the Table 1 was changed and the same corrections in the textwere made.
The sentence in Discussion section :
The history of CV event was observed in 66% of study subject and more frequently in the group of dependent patients (43% vs 23%) (Table 1).
was replaced with
The history of CV event was observed in 66% of study subject and more frequently in the group of dependent patients (89% vs 45%) (Table 1).
(8) The frequency of heart failure is 20.93% according to Table 3. However, the authors indicated the frequency as 27% in page 9, 3rd paragraph. The authors should revise the mistake.
and 89 % (55/62) in dependent patients. If so, the authors should revise the data in Table 1 and page 9, 3rd paragraph.
Response: One more time we sorry for the mistake. During writing the text we confused the number of patients with the percentage. This error was corrected accordingly.
The sentence in Discussionsection :
Heart failure as a complication of coronary heart disease was the most frequent cause of FD (in 27% of pts.)
was replaced with
Heart failure as a complication of coronary heart disease was the most frequent cause of FD (in 20,9% of pts.)
(9) There are some typographical errors in the manuscript. For example, join is not correct and joint is correct. FD is an abbreviation for functional disability but not functional decline (page 9, 3rd paragraph). The authors should correct the errors adequately.
Thank you for finding this errors. The text in Discussion section was adequately changed and abbreviation FD was concatenated with functional decline.
Additionally, the manuscript was corrected by professional English language editor employed in journal from the base of JCR - Journal Citation Reports.
Reviewer 3 Report
I have read with great interest for the work of Tomasz Gołębiowski and colleagues in which the authors analyze disability problems in hemodialysis patients in the context of cardiovascular complications and vascular conditions.
It is in fact known that cardiovascular health in the hemodialysis nephropathic patient is of primary importance.
Given the constant increase in the age of the population, it is very common for this category of fragile patients to come to the attention of specialists.
The study is rigorous and well written.
Alongside to the scales already used (Barthel Index), it would have been appropriate to classify patients according to the Rockwood scale, widely used to classify the independence of geriatric patients.
I suggest, if authors have the data, to provide information about aortic stenosis and possible, previous TAVR, that is frequently associated to CV comorbidities in these patients.
Author Response
I have read with great interest for the work of Tomasz Gołębiowski and colleagues in which the authors analyze disability problems in hemodialysis patients in the context of cardiovascular complications and vascular conditions.
It is in fact known that cardiovascular health in the hemodialysis nephropathic patient is of primary importance.
Given the constant increase in the age of the population, it is very common for this category of fragile patients to come to the attention of specialists.
The study is rigorous and well written.
Response: We thank the reviewer for the comment and evaluation of our study. We would like only point that many potential factors related to uremia (post dialysis fatigue, depression, cognitive impairment)in CKD patient may limit activity of daily leaving but our study shown that cardiovascular events in the past was the most important factor of functional decline.
Alongside to the scales already used (Barthel Index), it would have been appropriate to classify patients according to the Rockwood scale, widely used to classify the independence of geriatric patients.
Response: Thank you for underlying clinical usefulness of Clinical FRAILT Scale by Rockwood. We think that both scales are actually referring to activities of daily living. We used Barthel Index because it is our standard assessment in all Polish facilities, even nursing homes, with more linear scale (0-100 points). It allowed for better statistical analysis without data transformation for classes of Rockwood scale.
I suggest, if authors have the data, to provide information about aortic stenosis and possible, previous TAVR, that is frequently associated to CV comorbidities in these patients.
Response: In our cohort only 1 patient had aortic stenosis and underwent transcatheter aortic valve replacement (TAVR). We included this case in group of Heart failure/coronary heart disease/ acquired valvular heart disease in Table 3. Accordingly, in Table 3 of the revised version of the manuscript we added a phrase: acquired valvular heart disease.
Round 2
Reviewer 1 Report
- Regarding the definition of “cardiovascular (CV) events” in this manuscript, I don’t agree the CV contains ischemic/hemorrhagic stroke. In my understanding, the stroke is cerebrovascular disease, except for cardiogenic stroke.
Response: We understand this remark, however we used broad definition of cardiovascular diseases (CVDs) of Thomas, H.; Diamond, J.; Vieco, A.; Chaudhuri, S.; Shinnar, E.; Cromer, S.; Perel, P.; Mensah, G.A.; Narula, J.; Johnson, C.O., et al. Global Atlas of Cardiovascular Disease 2000-2016: The Path to Prevention and Control. Glob Heart 2018, 13, 143-163, doi:10.1016/j.gheart.2018.09.511
According to this guidelines CVDs include diseases of the heart, vascular diseases of the brain and diseases of blood vessels.
We added a reference and following sentence in Material and Methods section
“Cardiovascular (CV) diseases include diseases of the heart, vascular diseases of the brain and diseases of blood vessels [6] particularly:”
→ I accepted your kind response.
- What the authors says in this paper seems to be no different from “PWV was significant predictor of coronary heart disease, stroke and cardiovascular events, with hazard ratios 1.23, 1.28, 1.30, respectively. PWV is regarded as useful, non-invasive, diagnostic and prognostic tool for assessing cardiovascular disease, in the CKD population.”, which is cited in this article. In short, the CV events, that is predicted by PWV, causes the decline of ADLs, or the decrease of Barthel index. In my thought, it is an obvious thing, so the conclusions of this paper are nothing new.
Response: We thank you for this important comment. From our point of view CVDs end points (coronary heart disease, stroke etc.) are not similar with functional decline (FD). As we shown FD in CKD patients may be associated with many factors i.e. CV events, sarcopenia, depression, degenerative joint disease etc. Our study for the first time shown significant relationship between arterial stiffness surrogate, namely PWV and ADLs decline assessed by widely used in Poland Barthel Index. We could not find similar papers.
We also found that CV events in the past were more important culprit of FD than others factors related to uremia (post dialysis fatigue, sarcopenia, depression).
→
→Firstly, my comment was too strong. You are right, there are some new things, especially quite new study in Poland HD patients. As authors knows, there are several published paper relationship between data of PWV and CVD to predict prognosis of HD patients. For example, PLoS One. 2018, 13:e0206446, PLoS Comput Biol. 2018, 14:e1006417 Am J Nephrol. 2019, 49:317-327, and Hypertension. 2018, 71:1126-1132 as recent reports. The present results were similar to these results although there were some additional evaluated factors as small differences. If they could extend observational period to evaluate progression of disability, e.g. one year and two years after the first evaluation for each patients, and/or increase of number of samples, it may be improved.
Author Response
Regarding the definition of “cardiovascular (CV) events” in this manuscript, I don’t agree the CV contains ischemic/hemorrhagic stroke. In my understanding, the stroke is cerebrovascular disease, except for cardiogenic stroke.
Response: We understand this remark, however we used broad definition of cardiovascular diseases (CVDs) of Thomas, H.; Diamond, J.; Vieco, A.; Chaudhuri, S.; Shinnar, E.; Cromer, S.; Perel, P.; Mensah, G.A.; Narula, J.; Johnson, C.O., et al. Global Atlas of Cardiovascular Disease 2000-2016: The Path to Prevention and Control. Glob Heart 2018, 13, 143-163, doi:10.1016/j.gheart.2018.09.511
According to guidelines, CVDs include diseases of the heart, vascular diseases of the brain and diseases of blood vessels.
We have added a reference and following sentence in Material and Methods section.
“Cardiovascular (CV) diseases include diseases of the heart, vascular diseases of the brain and diseases of blood vessels [6] particularly:”
→ I accepted your kind response.
What the authors says in this paper seems to be no different from “PWV was significant predictor of coronary heart disease, stroke and cardiovascular events, with hazard ratios 1.23, 1.28, 1.30, respectively. PWV is regarded as useful, non-invasive, diagnostic and prognostic tool for assessing cardiovascular disease, in the CKD population.”, which is cited in this article. In short, the CV events, that is predicted by PWV, causes the decline of ADLs, or the decrease of Barthel index. In my thought, it is an obvious thing, so the conclusions of this paper are nothing new.
Response: Thank you for this important comment. From our point of view, CVDs end points (coronary heart disease, stroke etc.) are not similar with functional decline (FD). As we have shown, FD in CKD patients may be associated with many factors, i.e. CV events, sarcopenia, depression, degenerative joint disease, etc. Our study, for the first time, shows significant relationship between arterial stiffness surrogate, namely PWV, and ADLs decline, assessed by, widely used in Poland, Barthel Index. We could not find similar papers.
We also found that CV events in the past, were more important culprit of FD than other factors related to uremia (post dialysis fatigue, sarcopenia, depression).
→
→Firstly, my comment was too strong. You are right, there are some new things, especially quite new study in Poland HD patients. As authors knows, there are several published paper relationship between data of PWV and CVD to predict prognosis of HD patients. For example, PLoSOne. 2018, 13:e0206446, PLoSComput Biol. 2018, 14:e1006417 Am J Nephrol. 2019, 49:317-327, and Hypertension. 2018, 71:1126-1132 as recent reports. The present results were similar to these results although there were some additional evaluated factors as small differences. If they could extend observational period to evaluate progression of disability, e.g. one year and two years after the first evaluation for each patients and/or increase of number of samples, it may be improved
Response: Thank you for mentioning about this important reference. Naturally, many factors may influence measure results of pulse wave, for example: cardiac output, existence of AV fistula, hydration status and time of PWV assessment (before, during or post HD). This last issue was discussed with Reviewer 2 Round 1. To accommodate the Reviewer 1, we have included additional limitation in Discussion section and following sentence were added.
Precise data indicate that except stiffness of large and small arteries many factors may influence hemodynamic parameters of pulse wave analysis including; cardiac output, existence of arteriovenous fistula and hydration status of the individual patients [48].
To enrich the discussion, we have added this citation to our paper [Debowska, M.; Poleszczuk, J.; Dabrowski, W.; Wojcik-Zaluska, A.; Zaluska, W.; Waniewski, J. Impact of hemodialysis on cardiovascular system assessed by pulse wave analysis. PLoS One 2018, 13, e0206446, doi:10.1371/journal.pone.0206446.]
Although many publications indicate relationship between PWV and CVD and role in mortality prediction, you are right mentioning, other well validated risk score prediction model, like ARO-RS (Hypertension. 2018, 71:1126-1132). ARO-RS is not considering disability of patient and actually predict mortality risk for next years. As our study is cross-sectional, we have no data on long term survival as for now. Thanking for this remark we decided to extent follow up of out cohort for 3 years. We exchanged following sentence in Discussion section:
PWV is regardless a useful, non-invasive, diagnostic and prognostic tool for assessing cardiovascular disease in the CKD population
to
PWV is regardless a useful, non-invasive, diagnostic and prognostic tool for prediction cardiovascular disease in the CKD population however as mortality predictor may not necessary show superior usefulness comparing to other validated clinical risk scores. Recent comparison of PWV vs. ARO Risk Score (Annualized Rate of Occurrence) in patients with ESRD indicate that PWV has inferior prognostic power for all-cause and cardiovascular mortality to ARO Risk Score [21,22]. Additional data point that 24 h-PWV may have higher prediction value than PWV in the office setting for all-cause-mortality in hemodialysis patients [23]
Appropriate references were added.
- Tripepi, G.; Agharazii, M.; Pannier, B.; D'Arrigo, G.; Mallamaci, F.; Zoccali, C.; London, G. Pulse Wave Velocity and Prognosis in End-Stage Kidney Disease. Hypertension 2018, 71, 1126-1132, doi:10.1161/HYPERTENSIONAHA.118.10930.
- Anker, S.D.; Gillespie, I.A.; Eckardt, K.U.; Kronenberg, F.; Richards, S.; Drueke, T.B.; Stenvinkel, P.; Pisoni, R.L.; Robinson, B.M.; Marcelli, D., et al. Development and validation of cardiovascular risk scores for haemodialysis patients. Int J Cardiol 2016, 216, 68-77, doi:10.1016/j.ijcard.2016.04.151.
- Matschkal, J.; Mayer, C.C.; Sarafidis, P.A.; Lorenz, G.; Braunisch, M.C.; Guenthner, R.; Angermann, S.; Steubl, D.; Kemmner, S.; Bachmann, Q., et al. Comparison of 24-hour and Office Pulse Wave Velocity for Prediction of Mortality in Hemodialysis Patients. Am J Nephrol 2019, 49, 317-327, doi:10.1159/000499532.
Unfortunately, we have finished the study in December 2019, that is why we are not able to perform one or two year follow-up visits for evaluation progression of disability. We are grateful for interesting and valuable idea.
Reviewer 2 Report
The authors revised the manuscript adequately according to my concerns, except that English language is minor spell check required. For example, there are some typographical errors in joint. The authors should revise the mistakes..
Author Response
The authors revised the manuscript adequately according to my concerns, except that English language is minor spell check required. For example, there are some typographical errors in joint. The authors should revise the mistakes.
Response: We understand that last version of our manuscript included many changes (visible in track changes mode) that probably makes it difficult to read. Additionally we have problems with editor program which connect words automatically and it was probably associated with print character. We are sorry for that inconvenience. In revised manuscript we took off “track changes mode”, change the font type from Arial to Calibri and correct typographical errors. We hope that it fix the problem.